# Extracellular Vesicles as Novel Diagnostic and Therapeutic Agents for Non-Melanoma Skin Cancer: A Systematic Review

**DOI:** 10.3390/ijms25052617

**Published:** 2024-02-23

**Authors:** Konstantinos Seretis, Eleni Boptsi, Anastasia Boptsi

**Affiliations:** Department of Plastic Surgery, Medical School, University of Ioannina, 45100 Ioannina, Greece; elenabop53@gmail.com (E.B.); anastasiaboptsi@gmail.com (A.B.)

**Keywords:** non-melanoma skin cancer, basal cell carcinoma, cutaneous squamous cell carcinoma, exosomes, extracellular vesicles, biomarker, treatment, diagnosis, prognosis

## Abstract

Standard non-melanoma skin cancer (NMSC) treatment involves surgery, recently combined with chemotherapy or immunotherapy in cases of advanced tumors. EVs, including exosomes, are integral to carcinogenesis, and are found in NMSC releasing mediators impacting tumor progression. Nevertheless, the precise intercellular signaling role of NMSC-derived EVs remains unclear. This review aims to elucidate their potential role in NMSC diagnosis and treatment. This systematic review encompassed literature searches in electronic databases from inception to September 2023, based on certain inclusion and exclusion criteria, addressing NMSC-derived EVs, their molecular cargo, and their implications in the diagnosis, prognosis, and treatment of NMSC. Key components were identified. Extracellular vesicle (EV) proteins and RNA have emerged as diagnostic biomarkers in EV-based liquid biopsy. Circular RNA CYP24A1, known for its molecular stability, holds promise as a diagnostic biomarker. Long noncoding RNAs (lincRNA-PICSAR) and Desmoglein 2 (DSg2) are linked to drug resistance, serving as prognostic biomarkers. EV mediators are being actively investigated for their potential role as drug delivery agents. In conclusion, this systematic review showed that NMSC-derived EVs display promise as therapeutic targets and diagnostic biomarkers. Further research is imperative to fully comprehend EV mechanisms and explore their potential in cancer diagnosis and treatment.

## 1. Introduction

Non-melanoma skin cancers (NMSCs) represent approximately 30% of human cancers [1]. Numerous population-based studies have demonstrated that the incidence rates of the two main NMSC types, namely basal cell carcinoma (BCC) and cutaneous squamous cell carcinoma (cSCC), are rising [2]. The underlying pathogenesis of NMSC has not yet been fully elucidated, but a variety of factors, in addition to environmental exposure and UV radiation, are associated with an increased risk of developing NMSC [2]. Although the current available treatment options, including surgery and radiotherapy, are proven to be effective for the majority of NMSC cases, those at advanced stages and presenting metastatic tumors require systemic therapy, such as immunotherapy, epidermal growth factor receptor (EGFR) inhibitors, and platinum-based chemotherapy [1]. cSCC has a stronger malignant tendency to develop metastases or local recurrence [2], while BCC follows a less aggressive clinical progression; however, left untreated, both are related to significant morbidity [3]. Immunotherapy is currently considered the most effective option for unresectable NMSC [1]. The anti-PD1 inhibitors pembrolizumab and cemiplimab have been approved as the optimal treatment options for locally advanced and metastatic cSCC [1,4]. Cemiplimab has recently received FDA approval for advanced BCC, which was formerly treated with Sonic-Hedgehog inhibitors [1]. Even though immunotherapy appears promising, its potentially fatal adverse effects require careful selection of eligible patients [1]. In this context, other treatment options are researched.

Extracellular vesicles (EVs), such as exosomes, apoptotic bodies, microvesicles and oncosomes, have been reported as main determinants of the pathogenesis and progression of melanoma, and thus are currently explored as diagnostic and prognostic biomarkers [5]. Exosomes are small membranous vesicles (sEVs) with the ability to transfer their cargo among cells [6] and have a critical role in both physiological and pathological processes, such as carcinogenesis [3,7]. Exosomes are released by all cell types, including tumor cells, and widely exist in all body fluids such as blood, urine, saliva, cerebrospinal fluid, and amniotic fluid [6,8,9]. NMSC cell-derived exosomes generate and release mediators that modulate tumor growth and potential metastases [5]. However, the intercellular signaling role of EVs in NMSC is largely unknown [7].

The aim of this systematic review is to explore and summarize the current evidence regarding the possible role of EVs in NMSC, with a special focus on exosomes.

## 2. Methods

### 2.1. Study Protocol

A systematic review was conducted using a predetermined protocol established according to the Cochrane Handbook’s recommendations [10], registered in the PROSPERO database (registration number: CRD42023492207). This review adhered to the updated PRISMA (Preferred Reporting Items for Systematic Reviews and Meta-Analyses) guidelines, presented in the Appendix A [11].

### 2.2. Search Strategy

An electronic literature search in MEDLINE (PubMed), Scopus, the Cochrane Library, and CENTRAL electronic databases was conducted from inception to September 2023. No time and language restrictions were applied. The completed search strategy is included in the Appendix A. This search was supplemented by a review of reference lists of potentially eligible studies.

### 2.3. Eligibility of Relevant Studies

Studies met the following inclusion criteria: (1) prospective design; (2) evaluation of patients diagnosed with non-melanoma skin cancer, or cell lines from such populations; (3) reported data on components in extracellular vesicles with diagnostic, prognostic, or therapeutic value; and (4) publication in a peer-reviewed journal. We excluded studies reporting on mucosal SCC, on melanoma, or when the molecular cargo of EVs was not researched. Review articles, duplicate reports, and non-human studies were excluded as well.

### 2.4. Study Selection

Two reviewers (K.S. and E.B.) independently screened retrieved database files and the full script of potentially eligible studies for relevance. Disagreement was resolved by consensus.

### 2.5. Data Collection and Risk of Bias Assessment

Data extraction was conducted independently by the two reviewers using a standardized form. Discrepancies were resolved by consensus. We extracted data, including the general study characteristics, sample sources, incubation, experimental and analysis methods applied, target molecules studied, and outcomes of interest. The primary outcome considered was the functional effect of EVs on the target population of NMSC patients, which will enable their application as prognostic or diagnostic biomarkers or potential therapeutic targets.

The quality of studies was assessed using the SYRCLE’s tool specifically designed for in vivo research studies [12] and a customized Cochrane risk of bias tool (Appendix A) tailored to address the requirements of in vitro research [13]. The quality assessment of the included studies is available in the Appendix A.

### 2.6. Data Synthesis and Analysis

We provided a narrative summary of the included studies based on the EVs studied in the literature and the publication date. The outcomes presented were further categorized according to the functional role of the EVs as prognostic, diagnostic, or therapeutic targets.

## 3. Results

A PRISMA flowchart of the included studies is presented in Figure 1. Through the applied search strategy, 888 articles were retrieved, of which 245 duplicates were removed. The remaining articles were screened based on title and abstract, resulting in 16 articles being sought for retrieval. For one article, there was only an abstract available; thus, 15 full texts were eventually assessed for eligibility. Based on the inclusion criteria, eight studies were included in this systematic review.

### 3.1. General Study Characteristics

The studies included were conducted in the USA (three), China (three), and Austria (one). The eighth study included was a multicenter study involving centers in Austria, Japan, UK, Chile, and the USA. All of the studies were published after 2017. Study characteristics are summarized in Table 1. All of the studies were prospective clinical (five) or experimental (three) studies. Among the eight studies, five analyzed patient samples, while three analyzed cell lines. Patient samples were sourced either from serum or tissues, including skin from the healthy controls. All studies followed a sequential ultracentrifugation protocol for EV isolation, and EVs were characterized using Transmission Electron Microscopy (TEM), Nanoparticle Tracking Analysis (NTA), and Western blotting for exosomal markers (CD81, CD9, and CD63).

The role of EVs as prognostic biomarkers, diagnostic biomarkers, or therapeutic agents was examined in two, three, and six studies, respectively (Table 1).

### 3.2. EV Structure and Function

Exosomes are nano-sized EVs, which differ in morphology, biological properties, biogenesis, and functional roles from other larger types of EVs [4,6,8,20]. Both exosomes and other EVs carry and deliver information through biologically active molecules and mediators and regulate skin homeostasis or disease pathogenesis [8,21]. A phospholipid bilayer with protein markers on the surface, dependent on each cell function, forms their membrane [8,20]. These protein markers can be used to differentiate tumor-derived exosomes [6].

Depending on their intracellular origin, EVs can be categorized as exosomes or microvesicles [21]. Exosome biogenesis follows certain stages. After the infolding of the cytoplasmic membrane generates early endosomes, they integrate molecules, such as DNA, RNA, proteins, and lipids, into multi-vesicular bodies (MVBs). The fusion of MVBs with the cytoplasmic membrane leads to the release of exosomes into the extracellular space [20]. On the other hand, microvesicles are formed through direct budding from the plasma membrane [21]. The size of exosomes typically ranges between 40 and 120 nm, while microvesicles vary more in size, from 50 to 1000 nm [22]. Due to the size overlap and the lack of specific markers, distinguishing between the two types is difficult without observing the formation process. However, determining the biogenesis pathway of EVs remains challenging, except when live imaging techniques are employed [23]. In this review, we adhere to the original authors’ definitions of exosomes.

EVs, including exosomes, are characterized by carrying a load of biologically active molecules and mediators. These include lipids, proteins, amino acids, metabolites, and nucleic acids, notably microRNA (miRNA), non-coding RNAs (ncRNAs), long non-coding RNAs (lncRNA), and circular RNAs (circRNA) [8,21]. Through the release of these components, EVs enable intercellular communication between both cancer and normal cells, via direct contact or receptors [5,20]. As a result, specific genetic information is transferred from cancer to benign cells, contributing to cancer progression and metastasis [8].

### 3.3. EVs Role in Tumorigenesis

Tumor cells release a large quantity of EVs that are critically involved in cancer pathogenesis, progression, and metastasis [5,8,21]. Tumor-derived exosomes (TEXs or TDEs) mediate between tumor cells and the tumor microenvironment (TME) [16] and can induce tumor progression and even metastasis by supporting tumor initiation, angiogenesis, epithelial-to-mesenchymal transition (EMT), matrix remodeling, and immune modulation [9]. These processes are modulated largely by the cargo of miRNAs transported via exosomes. Specifically, the overexpression of miRNAs induces cancer development, whilst miRNA underexpression is related to an inability to suppress the expansionary tendency of tumors [18,21].

Overmiller et al. also demonstrated that SCC-derived EVs can alter the TME by inducing the proliferation of local fibroblasts [7]. TDEs are also implicated in both pro- and anti-tumor immune responses. Immune escape is induced by modulating the expression of IL-6, leading to deregulation of dendritic cell maturation. Dendritic cells are an essential immune system cell population. The anti-tumor defense mediated by natural killer cells (NK) is also activated by IL-6 expression modulation. NK cells, stimulated by TDEs and the pro-inflammatory cytokines that are secreted, induce tumor cell apoptosis [5]. On the other hand, TDEs can also decrease NK cells within the TME and thus induce immune suppression [5].

### 3.4. EVs as Therapeutic Targets in cSCC

In view of the role of EVs in tumorigenesis and metastasis, EVs and their cargo could be potential therapeutic targets. In fact, in head and neck SCC (HNSCC), Teng et al. hypothesized that targeting exosomes derived from irradiated HNSCC cells would downregulate the AKT pathway, and thus radiosensitivity could be enhanced. They also reported the ability of many exosome inhibitors such as GW4869 to disrupt the lipid composition, indomethacin, or Ras inhibitors [6].

#### 3.4.1. Circ-CYP24A1

A potential target that has recently gained attention due to its specificity and stability both as a diagnostic biomarker and therapeutic target is circRNA, a non-coding RNA whose expression is enriched and stable in exosomes. It is postulated that there is an association between circRNAs and the development of multiple human tumors, such as gastric cancer. CircRNAs can induce the progression and peritoneal metastasis of gastric cancer by translocation to the target cells via exosome secretion [24]. Zhang et al. deployed RNA sequences to form expression profiles of exosomal circRNAs in cSCC. They compared the plasma exosomes derived from five cSCC patients with five healthy samples, showing an upregulation of 25 and a downregulation of 76 circRNAs in cSCC, among the 7577 differentially expressed circRNAs [3]. Upregulated circRNAs, such as circ-CYP24A1, were principally involved in the immune response, while downregulated circRNAs were modulators in metabolic pathways in cancer cells and RNA transportation. A correlation between the tumor’s clinical characteristics and circ-CYP24A1 was also reported. The inhibition of exosomal circ-CYP24A1 was shown to possibly also affect SCC progression by suppressing the tumor’s locally invasive and metastatic dynamic [3]. These data indicated that the exosomal circ-CYP24A1, among other exosomal circRNAs, might be considered a potential therapeutic target to restrain the development, migration, and invasion of cSCC [3].

#### 3.4.2. Desmoglein 2 (Dsg2)

Another key target component seems to be desmoglein 2 (Dsg2), a desmosomal cadherin. Dsg2 is often overexpressed in cancers, including NMSC, and is associated with poor prognosis in melanoma as it promotes tumor angiogenesis [18]. Flemming et al. and Overmiller et al. reported that Dsg2 is implicated in the release of EVs from SCC keratinocytes enriched with cytokines, such as IL-8, and is impoverished in miR-146a [7,18]. The active role of Dsg2 in the biogenesis of EVs was confirmed by Overmiller et al. when the overexpression of Dsg2 both in non-cancerous and SCC cells was compounded with enhanced EV secretion. In A431 SCC cells the secretion of EVs was also decreased when Dsg2 was targeted with shRNA [7]. Through the release of EVs, Dsg2 also promoted tumor proliferation in both cutaneous and head and neck SCCs. To explore the underlying mechanisms, the release of cytokines, which are known for their role in tumor growth, from the exosomes in response to increased levels of Dsg2 was investigated. It was reported that IL-8, which also induces tumor progression and immune response, was significantly increased. Considering the downregulation of miR-146a in those cells, the enhanced expression of the IL-8 gene was the outcome of the unsuccessful inhibition of the NFkB signaling pathway by miR-146a. As a result, the lower levels of miR-146a led to the increased expression of IL-8 [18]. Moreover, the correlation between immunotherapy response and IL-8 rate was explored. Following the measurement of IL-8 levels in patients with HNSCC under therapy with nivolumab, an anti-PD1 agent, treatment response rates were found to be higher in patients with significantly lower expression of IL-8 [18]. Even though more research is required to elucidate the implementation of those ascertainments in relation to cSCC, targeting Dsg2 or IL-8 could enhance the susceptibility of SCC cells to immune agents and ameliorate provided therapy.

Overmiller et al. demonstrated that SCC-derived EVs are enriched with Dsg2-C-terminal fragment (Dsg2-CTF), which occurs after the modification of the full length Dsg2 by metalloproteinase 17 (ADAM17). Since ADAM17 levels are increased in cSCC, they hypothesized that during malignant transformation, Dsg2 fractures into a ~95 KDa ectodomain and intracellular CTF, which has an essential role in EV secretion in SCC cells. In view of this fact, post-translational Dsg2 alteration seems promising as a research field for new treatment strategies [7,25]. Flemming et al. also suggested that besides Dsg2 proteolysis, the palmitoylation of Dsg2 is essential for the release of sEVs, which rendered palmitoylacyltransferases (PATs) as possible therapeutic targets [18].

#### 3.4.3. p38 Inhibited Cutaneous Squamous Cell Carcinoma-Associated lincRNA (PICSAR)

Following the current guidelines for SCC patients to minimize the risk of metastasis or recurrence, surgery is accompanied by radiotherapy, and less often chemotherapy. However, the acquired drug resistance of tumor cells represents a significant impediment to appropriate therapy provision. Extracellular vesicles, secreted from tumor cells, facilitate cancer cell adaptation to microenvironmental conditions and chemoresistance through the transfer of ncRNAs [5,17]. Long noncoding RNAs (lncRNAs), namely noncoding RNAs with >200 nucleotides, participate in the regulation of multiple human cancers, while their deregulation is associated with chemoresistance [17]. Wang et al. investigated the role of PICSAR in the resistance of cSCC cells to cisplatin (DDP), a chemotherapeutic drug commonly used in cSCC treatment. Lnc-PICSAR was elevated in the exosomes derived from SCC patients’ serum and SCC cells compared to non-cancerous cells. In addition, lnc-PICSAR levels were higher in DDP-resistant SCC cells than in DDP-sensitive cells [17]. The correlation between lnc-PICSAR and miR-485-5p and REV3L was also studied. Lnc-PICSAR is involved in the regulation of SCC chemoresistance by inhibiting miR-485-5p, which subsequently promotes the expression of REV3L. Based on these data, exosome-mediated lnc-PICSAR could present a potential prognostic biomarker to evaluate the treatment response, as well as a therapeutic target [17].

#### 3.4.4. miRNA

Recent studies have investigated the role of miRNA in the metastatic potential of BCC, melanoma, breast, prostate, and lung cancer [26,27,28,29]. Chang et al. isolated exosomal miRNA from patients with metastatic BCC (MBCC) and non-metastatic BCC (non-MBCC) and reported that exosomes in patients with MBCC increased the proliferation and invasion ability of fibroblasts [14]. Among the isolated miRNAs, nine were significantly overexpressed in MBCC in comparison to non-MBCC. The role of mir-197 was further investigated, considering its role in non-BCC tumors. Even though mir-197 was found to be present at enhanced levels in patients with MBCC, its inhibition was not correlated to decreased fibroblast and keratinocyte proliferation [14].

### 3.5. EVs as Diagnostic Biomarkers

The stability and the easy collection of EVs from the circulation and body fluids through non- or minimally invasive methods are attractive features of exosomes, demonstrating their potential role as biomarkers of different diseases [6,8]. Although research on the therapeutic possibilities of EVs is at an early stage, their diagnostic role has been already explored by many recent studies that focus on the value and utility of EV-based liquid biopsy. It was reported that an EV-protein and RNAs are effective biomarkers for stage I and II pancreatic cancer screening, achieving excellent rates of both specificity and sensitivity when combined [30]. Similar research has also demonstrated the utility of an EV-RNA for the diagnosis of non-small lung cancer [25,31]. In HNSCC, liquid biopsy recognizes exosomal miRNAs and exosome-derived proteins [6]. In view of the growing demand for early screening and diagnosis, research on cSCC has also turned towards the identification of biomarkers.

Sun et al. studied the expression of Ct-SLCO1B3, an EV tumor marker gene, in patients with recessive dystrophic epidermolysis bullosa (RDEB). It was demonstrated that the gene was expressed only in RDEB-SCC-derived EVs, and thus could be considered a potential diagnostic biomarker, with the perspective that more studies will be conducted to ascertain whether those results apply to the general population [15]. Moreover, Zauner et al. demonstrated that there is a specific miRNA panel that can distinguish RDEB-cSCC from RDEB lesions and healthy skin samples, and thus can be used as a diagnostic biomarker. Based on those findings, they proposed a tumor detection model. However, due to the small available sample, they used miRNA-seq panels of HN-SCC as supplementary data, since the miRNA profiles of the two cancers displayed significant similarities. As a result, three tumor detection models were created which included 33, 10, and three miRNAs that were significantly deregulated in HN-SCC and RDEB-cSCC exosomes. Each model was tested on the HN-SCC training set and then its predictive ability was evaluated both on the HN-SCC set and RDEB-exosome data. They demonstrated that the less complex model that was based on three unique miRNAs could accurately predict tumors. However, clinical research is required to assess the applicability of this model [19].

As mentioned above, besides their role as a therapeutic target, circ-CYP24A1 and linc-PICSAR can serve as diagnostic biomarkers as well. Notably, circ-CYP24A1 is considered an excellent diagnostic biomarker, mostly because of its resistance to the catalytic effect of RNAse [3].

### 3.6. EVs’ Role in Prognosis

The possibility of biomarkers used to strategically evaluate cancer treatment effects was also explored [8]. In HNSCC, Theodoraki et al. studied the role of circulating exosomes as biomarkers of completely cured or relapsed disease by isolating exosomes from different stages of the treatment timespan: before, during, and after therapy [32]. Regarding cSCC, linc-PICSAR was linked to enhanced chemoresistance, and Dsg2 was hypothesized to participate in causing a decreased immunotherapy response [17,18].

### 3.7. EVs as Drug Delivery Systems

EVs, especially exosomes, are also being researched as potential drug delivery agents [8]. Up to this time, drug delivery systems have included peptides, polymers, nanoparticles, liposomes, and vector viruses. However, several issues have emerged regarding the imminent immune reaction to foreign molecules and the questionable success of perfusion into the target cell population. On the other hand, EVs have certain characteristics that render them ideal candidates for drug delivery. In order to fulfill their role as cell-to-cell mediators, the phospholipid bilayer of EVs offers resistance to external degrading forces of the circulatory system, as to protect their molecular cargo, leading to longer circulating half-life [21,33]. Moreover, EVs can infiltrate the blood-brain barrier, thus expanding their target group. Last but not least, since EVs are autologous mediators, the immune response is not induced [21,22]. EVs derived from cancer cells can be used as an excellent drug delivery system not only for chemotherapeutic drugs but for miRNA-based gene therapy, since EVs naturally carry miRNA. Due to increasing interest, the usage of EVs as natural drug carriers has been investigated in many cancer types, among which are HNSCC, breast, colon, gastric, and brain cancers [21,34,35,36]. However, there is still little to no research on whether this therapy can be applied for cSCC patients.

### 3.8. Prospects for EVs in NMSC Anticancer Therapy: The Promising Role of Exosomes

NMSC has historically been treated with surgical excision combined with chemotherapy or more recently with immunotherapy. Current progress in the field of molecular biology regarding the role of EVs and their molecular or protein cargo in cSCC has made the identification and use of EVs as therapeutic targets appealing, and maybe feasible. EVs as a drug delivery system seem very promising. More isolation methods and novel loading techniques have been developed to meet increasing demand. EVs could be isolated from the patient’s own cells and be appropriately modulated with specific markers to target specific cancer cells. In fact, clinical trials with plant and human tissue-derived exosomes have been carried out for melanoma, lung, and colorectal cancer [6]. Overmiller et al. reported that co-cultivating fibroblasts with A431-Dsg2/GFP cells led to the modification of a higher percentage of fibroblasts into GFP+ ones compared to cultivation with A431-GFP cells. These results indicate that the invasion of EVs into adjacent cells is facilitated by Dsg2, and thus EV Dsg2 modification could enhance drug delivery to cSCC cells. Moreover, as mentioned, Dsg2 post-translation modification is a promising therapeutic target. miRNA alteration could be used to target the processed Dsg2, given its importance in gene expression and its reported anti-glioma and anti-hepatocellular carcinoma activity [25].

However, there are still unmet challenges. Currently, ultracentrifugation is the gold standard isolation method. However, the low yield and the risk of EV degradation due to the strong forces applied demand further evolution of the existing techniques [33]. By the time purified EVs can be successfully isolated, the field of miRNA-based therapy should evolve as well. Besides the proper isolation of EVs, more hurdles are yet to be overcome, such as the loading of modulated miRNA without damaging the EV membrane and thus leading to the alteration of its desired properties [21]. Moreover, due to their low bioavailability, the prospect of using synthetic mimics has also been explored [6].

Zhao et al. reported a novel anti-tumor therapy for premalignant and malignant skin lesions [16]. Photodynamic therapy with 5-aminolevulinic acid (ALA-PDT) consists of a photosensitizer and light sources, which may lead to vascular endothelial cell injury, tumor cell apoptosis, and suppression of tumor metastasis or relapse [16]. The authors reported that the anti-tumor properties of exosomes are mediated by the maturation of dendritic cells (DC) and fibroblast secretion of TGF-β1. Dendritic cells are known for their antigen-presenting property, which induces the immune response. Subsequently, they demonstrated that matured DCs increased the secretion of IL-12, which activated cytotoxic T lymphocytes (CTLs) and resulted in the inhibition of tumor growth. In this study, ALA-PDT exosomes were shown to potentially stimulate the maturation of dendritic cells (DC), which are identified in the TME. Therefore, they demonstrated that ALA-PDT exosome therapy can lead to an anti-tumor immune response [16].

## 4. Discussion

While non-melanoma skin cancers are often uniformly regarded as non-aggressive tumors, this is true only for small lesions, treated early and adequately. Both BCCs and cSCCs are characterized locally by variable destructive growth and invasion of the surrounding tissues. Regional tissue invasion occurs less often, via lymph node metastasis, and distant invasion has a low rate of metastasis in the case of cSCCs [37,38,39]. Although the mortality rate associated with these tumors is relatively low, they pose a substantial health burden globally due to their high prevalence. In addition, they are linked to significant morbidity, particularly when affecting cosmetically sensitive areas such as the face, often necessitating challenging reconstructive operations [40,41,42,43].

Failure to resect a NMSC completely predisposes the disease to recurrence, stimulates the cancer cells’ aggressiveness, and increases the metastatic potential of SCC [37,38,39]. Early detection and intervention are thus crucial to minimizing the potential consequences of NMSC, emphasizing the importance of effective prognostic and diagnostic biomarkers in mitigating the impact of NMSC. The identification of tumor-derived EVs and their validation as biomarkers holds promise in refining risk stratification, driving guidelines-based treatment decisions and enhancing overall management strategies for individuals diagnosed with NMSC. Prognostic biomarkers can offer insights into the likelihood of cSCC progression and its potential for metastasis, aiding clinicians in tailoring surgical and oncological interventions to the specific needs of patients. Diagnostic biomarkers, on the other hand, play a crucial role in early detection, allowing for timely and targeted therapeutic management. Research efforts are focused on unraveling the molecular and genetic signatures associated with these tumors, aiming to identify molecular biomarkers that can be reliably assessed through non-invasive methods. In 2022, Lee et al. reviewed the literature on the genomics, transcriptomics, and proteomics of SCC-derived extracellular vesicles, encompassing pre-clinical and clinical studies [18]. Certain molecules were shown to be involved in EV-mediated tumor invasion and drug resistance, thus serving as potential prognostic and therapeutic predictors.

In cases of locally advanced or metastatic BCC where surgery and radiotherapy are not optimal treatment options, sonidegib and vismodegib are considered the standard treatment. These agents are hedgehog pathway inhibitors (HhIs) with a comparable efficacy and risk of adverse events. Despite their different pharmacokinetics, no significant clinical relevance has been demonstrated [44]. For cSCC, cemiplimab, a monoclonal anti-PD-1 antibody, has emerged as an alternative therapeutic option. Cemiplimab was also investigated as a potential treatment for BCC resistant in therapy with HhIs, receiving FDA approval for this indication. Nevertheless, the selection of immunotherapy options requires cautious evaluation of the risks to benefits profile, since these drugs can lead to severe, and even fatal, adverse effects [1]. Moreover, the issue of drug resistance also presents a challenge, necessitating research on novel targeted therapies [45].

This systematic review focused on the collection, analysis, and synthesis of the available evidence, highlighting advances in our understanding of the intricate molecular pathways involved in NMSC, and the potential role of tumor-derived EVs as therapeutic targets, especially in metastatic BCCs and advanced SCCs. One of the strengths of this review is the rigorous methodology applied, limiting the risk of bias, and therefore enhancing the presented outcomes of interest. In addition, the utilization of a tool to grade the confidence of the reported results further improved the conducted study analysis.

Nevertheless, this review is still subject to limitations. Of note is the relatively small number of included studies, an inherent drawback of a cutting-edge research field. The different molecular pathways involved in BCC and SCC tumorigenesis also increase the heterogeneity, although the outcomes associated with these tumors were presented separately.

In the light of MISEV 2018 and MISEV 2022 guidelines, the use of the term “exosomes” is discouraged due to lack of consensus on exclusive biomarkers to characterize the EV subtypes. This term should be used very cautiously, only if the subcellular origin is experimentally proved, which is also complicated by overlapping of the size and biomarkers in different EV subtypes [23,46]. As a review, the primary objective here was to summarize and analyze the existing literature rather than identify exosomes according to the strict criteria described in the MISEV guidelines. Therefore, this review has referred to EVs with the terminology used in the included studies to maintain the integrity of their findings. However, according to the guidelines, it is recommended to use terms for EV subtypes based on physical characteristics, such as size and density, biochemical composition, or cell of origin. This approach improves clarity and uniformity across studies, avoiding the use of potentially ambiguous and poorly defined terms.

Overall, further research is anticipated to shed light in this evolving landscape of NMSC molecular mechanisms, enabling the integration of biomarkers into clinical practice in order to improve the diagnostic accuracy and the predictive ability of each NMSC clinical course, as well as the availability and efficacy of targeted therapeutic options, even in cases of advanced tumors. Moreover, acknowledging the importance of uniform and precise terminology in enhancing the transparency of scientific outcomes, future studies should adhere to the current guidelines for EV characterization.

## 5. Conclusions

The expanding knowledge of the functions and possibilities of EVs is intriguing. As illustrated in this review, EVs, specifically those identified as exosomes, have an essential role in the pathogenesis and metastasis of cSCC via the induction of microenvironmental changes and cellular interactions between cancerous and benign cells. An increasing number of studies have focused on EV-based therapy, reporting potential therapeutic targets including circ-CYP24A1, lnc-PICSAR, translated Dsg2, and miRNAs. However, more research is warranted to elucidate the clinical applicability of such findings.

## Figures and Tables

**Figure 1 ijms-25-02617-f001:**
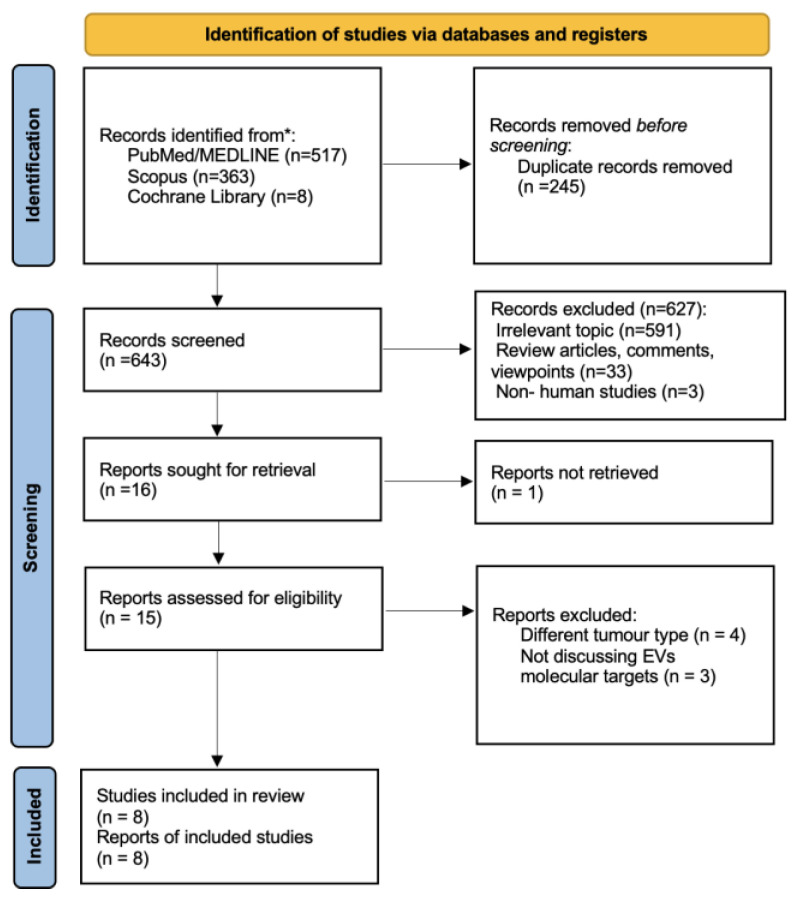
PRISMA flowchart for study selection.

**Table 1 ijms-25-02617-t001:** Characteristics of the studies included in this review.

Author, Year	Target Molecule	Source	Study Group	Control Group	Incubation	Exp. Method	Analysis Method	Outcome	Function
Overmiller A. et al. (2017) [7]	Dsg-2-CTF	Cell lines	-	-	A431/GFP, A431-Dsg2/GFP, HaCaT/GFP, HaCaT-Dsg2/GFP, primary NHK	In vitro	Western blot	Overexpression in SCC-derived EVs resulted in increased EVs secretion; inhibition of proteolysis of Dsg2 resulted in reduced EVs secretion	Therapeutic target
Chang et al. (2017) [14]	miR197	Serum	9 MBCC patients	9 non MBCC patients	NHK, human skin fibroblasts	In vitro	PCR	SS upregulation in MBCC patients; no impact on proliferation noted in fibroblasts and keratinocytes	Therapeutic target, prognostic biomarker
Sun et al. (2018) [15]	Ct-SLCO1B3	Tissue	RDEB-SCC patients	-	RDEB, RDEB-SCC, NHK	In vitro and in vivo	PCR	Expression of Ct-SLCO1B3 only in RDEB-SCC derived EVs	Diagnostic biomarker
Zhao Z. et al. (2020) [16]	ALA-PDT exosomes	Cell lines	-	-	SCCs (human A431, mouse PECA, primary mice SCCs), fibroblasts 3T3, DCs	In vitro	Western blot	Stimulation of DCs maturation and fibroblasts’ TGF-β1 secretion, leading to an anti-tumor immune response	Treatment
Wang et al. (2020) [17]	lnc-PICSAR	Serum	30 cSCC patients	30 healthy patients	NHEK, A431, HSC-5 cells	In vitro and in vivo	PCR	Elevated in cSCC cells and DDP-resistant cSCC cells	Prognostic biomarker, therapeutic target
Flemming J. et al. (2020) [18]	Dsg-2	Cell lines	-	-	A431/GFP, A431-Dsg2/GFP, A432-Dsg2cacs/GFP	In vitro and in vivo	Western blot	Inhibited palmitoylation of Dsg-2 corellates with reduced sEVs secretion and attenuated tumor development	Therapeutic target
Zhang Z. et al. (2021) [3]	circ-CYP24A1	Serum	5 cSCC patients	5 healty patients	A431, SCL-1 cells	In vitro	PCR	Upregulated in cSCC EVs; inhibition leads to attenuation of the tumor’s metastatic dynamic	Therapeutic target, diagnostic biomarker
Zauner R et al. (2023) [19]	miRNA (expr. profile)	Tissue	6 RDEB-cSCC	4 healthy patients, 5 RDEB	-	In vitro	PCR	51 miRNAS found significantly up-regulated and 74 down-regulated in RDEB-cSCC compared to RDEB	Diagnostic biomarker

Expr.: expression, Exp: experimental, SS: statistically significant, Dsg2: desmoglein 2, CTF: C terminal fragment, A431: epidermoid carcinoma cell, GFP: green fluorescence protein, HaCaT: normal keratinocytes, NHK: normal human keratinocytes, cSCC: cutaneous squamous cell carcinoma, EVs: extracellular vesicles, MBCC: metastatic basal cell carcinoma, RDEB: recessive dystrophic epidermolysis bullosa, ALA-PDT: 5-aminolevulinic acid photodynamic therapy, DCs: dendritic cells, NHEKs: Normal human epidermal keratinocytes, DDP: cisplatin, HSC-5: human skin squamous cell carcinoma, lnc-PICSAR: long noncoding RNA p38 inhibited cutaneous squamous cell carcinoma-associated lincRNA, Dsg2cacs: unpalmitoylated Dsg2, sEVs: small extracellular vesicles.

## Data Availability

No new data were created or analyzed in this study. Data sharing is not applicable to this article.

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
