# Peer review of "Extracellular Vesicles as Novel Diagnostic and Therapeutic Agents for Non-Melanoma Skin Cancer: A Systematic Review"

_ijms, 2024, doi:10.3390/ijms25052617_

Round 1

Reviewer 1 Report

Comments and Suggestions for Authors

This manuscript presents an excellent and comprehensive review of the role of exosomes in the diagnosis and therapy of non-melanoma skin cancer. The authors should be commended for their contribution to a field that is both innovative and currently under-represented in research literature. The manuscript is well-structured, thoroughly researched, and provides valuable insights into the potential of exosomes as novel agents in the medical field.

The manuscript is of high quality and can be accepted in its current form. However, there are some minor suggestions that could further enhance the clarity and impact of the paper:

  1. In line 32, before the word "elucidated," the addition of "fully" may provide more emphasis and clarity to the statement, highlighting the depth of the review.

  2. The legend in Table 1 (lines 126-133) could be improved for better clarity. Ensuring that the glossary of abbreviations is clearly indicated as referring to Table 1 would help readers in understanding and referencing the table more efficiently.

  3. The statements in lines 313-314 and 329-328, which refer to the same phenomenon, could be condensed for brevity and to avoid redundancy. This would streamline the text and maintain the focus of the readers.

  4. The heading of subsection 2.5 could be moved to line 298. This change would align the section heading more accurately with the content, as the discussion on miRNAs concludes at line 297.

Overall, the manuscript is a significant contribution to the field, offering a detailed overview of current research and potential future directions in the use of exosomes for non-melanoma skin cancer. The proposed minor revisions are optional and do not detract from the overall quality and value of the work. The authors have successfully navigated the complex topic of exosomes in cancer therapy and diagnosis, providing a clear, concise, and informative resource that is likely to be of great interest to researchers and clinicians in this field. The systematic approach to the review, coupled with the thoughtful analysis and discussion of the findings, makes this manuscript a valuable addition to the existing body of literature on non-melanoma skin cancer and exosome research.

Author Response

Thank you for reviewing our manuscript and your kind words.

  1. In line 32, before the word "elucidated," the addition of "fully" may provide more emphasis and clarity to the statement, highlighting the depth of the review.Thank you for your comment, this word was added.

  2. The legend in Table 1 (lines 126-133) could be improved for better clarity. Ensuring that the glossary of abbreviations is clearly indicated as referring to Table 1 would help readers in understanding and referencing the table more efficiently.

          The legend was amended and the abbreviations were checked. 

3. The statements in lines 313-314 and 329-328, which refer to the same phenomenon, could be condensed for brevity and to avoid redundancy. This would streamline the text and maintain the focus of the readers.

Thank you for pointing this out. The sentence in lines 328-29 was deleted.

4. The heading of subsection 2.5 could be moved to line 298. This change would align the section heading more accurately with the content, as the discussion on miRNAs concludes at line 297.

Thank you for your comment. The structure was amended, including specific subsections for exosomes' role in diagnosis, prognosis, and as drug delivery system, to ease reading and understanding of the review contents.

Reviewer 2 Report

Comments and Suggestions for Authors

A systematic review, submitted to IJMS by Konstantinos Seretis and the co-authors, is devoted to the role of exosomes (EVs) as diagnostic and therapeutic agents for skin cancer.

Several significant issues do not allow the reviewer to recommend the paper to be accepted even after major revision. The article should be redesigned and cannot be published in its present form.

1) The most significant issue concerns the use of terms.

As seen in Section S2, the authors used several keywords in the meta-analysis, including extracellular vesicles, as well as Exosomes, Microvesicles, Small Extracellular Vesicles, and even Small Vesicles. The title of the article includes “exosomes,” although this is only a particular fraction of extracellular vesicles. Moreover, according to the MISEV2018 (and other) guidelines, the term "Exosomes" is not recommended when referring to EVs. Moreover, what reasons do the authors have for using a more specific term instead of a more general one? Did the authors check the compliance of the EVs in the articles with the MISEV requirements for exosomes? Since a huge part of the publications call any fraction of EVs exosomes, it is extremely important to use the terms carefully.

2) Unfortunately, the authors do not pay attention to the physiological significance of exosome markers, which have been shown to be associated with cancer pathology.

3) Based on the title, it would be logical to see a section dedicated to the diagnostic use of exosomes (EVs)

4) The boundaries of Table 1 extend beyond the page. What is the point of indicating the author of the article and the year in the first column?

5) Subsections of section 3 are numbered as 2.1, etc.

After eliminating these issues and thoroughly rewriting and redesigning the, it might be resubmitted.

Author Response

Thank you for reviewing our manuscript and the valuable comments.

1) The most significant issue concerns the use of terms.

As seen in Section S2, the authors used several keywords in the meta-analysis, including extracellular vesicles, as well as Exosomes, Microvesicles, Small Extracellular Vesicles, and even Small Vesicles. The title of the article includes “exosomes,” although this is only a particular fraction of extracellular vesicles. Moreover, according to the MISEV2018 (and other) guidelines, the term "Exosomes" is not recommended when referring to EVs. Moreover, what reasons do the authors have for using a more specific term instead of a more general one? Did the authors check the compliance of the EVs in the articles with the MISEV requirements for exosomes? Since a huge part of the publications call any fraction of EVs exosomes, it is extremely important to use the terms carefully.

We recognize the critical importance of adhering to MISEV guidelines to ensure clarity and accuracy in the EV research. In addressing your question, we carefully considered the use of specific markers in the studies included. There is indeed no consensus on exclusive exosomal markers or specific size range to use as identification for exosomes, thus live imaging techniques are recommended as a reliable way to characterize exosomes. However, we noted that several studies included in our review identify EVs as exosomes based on the presence of the tetraspanins CD63, CD9 and CD81, despite the remark of MISEV guidelines. Therefore, regarding the compliance of EVs with MISEV requirements for exosomes, we recognize that not all of the studies explicitly meet the criteria set for defining EVs as exosomes. However, the primary objective of this review was to summarize the outcomes of the existing literature and shed light in the evolving standards and the difficulties in establishing subcellular origin with certainty. It should be noted that future studies should follow the EVs isolation and characterization methods proposed by MISEV guidelines and cautiously use the terminology to enhance the reproducibility and reliability of research findings.

Regarding our search strategy, the decision to include a wide range of terms was driven by the desire to not miss any relevant study that would not strictly adhere to the latest terminological guidelines. This approach deemed necessary due to the observed variability in the term usage, across literature. Also, our title specifically mentions “exosomes” to reflect the common usage of the term in the literature and the focus of most of the studies included in our review. We acknowledge the confusion and the importance of a clear and focused search strategy, but the range of the terms was chosen to ensure that our review is as inclusive as possible. 

We have included a section in our discussion to address the complexities and ongoing debates regarding the terminology used for EVs subtypes (425-441). In this way, we aim to present the current state of consensus and emphasize the need for consistent terminology in advancing EV research. Furthermore, we described in the results part the main techniques employed by the included studies for EV isolation and characterization, including the biomarkers that were considered indicative of exosomal origin (lines 122-125). The aim was to provide a clear overview of the criteria and methodologies applied in the included studies. 

2) Unfortunately, the authors do not pay attention to the physiological significance of exosome markers, which have been shown to be associated with cancer pathology.

Thank you for your comment.

This review is dedicated to the association of exosomes with NMSC. In this revision, A. we have added the main techniques employed for EV isolation and characterization, including the biomarkers that were considered indicative of exosomal origin. B. we have inserted subsections of the role of exosomes as diagnostic and prognostic biomarkers, in the Results and C. a paragraph, regarding hedgehog pathway and PD-1 inhibitors.

3) Based on the title, it would be logical to see a section dedicated to the diagnostic use of exosomes (EVs)

We appreciate your observation regarding the expected section on the diagnostic use of exosomes (EVs). Upon reevaluation, we realized that during the process of transferring our draft into the final manuscript template, the section adhering to the diagnostic use of exosomes was omitted. This was an oversight on our part. 

The studies adhering to the diagnostic role of exosomes are included in a separate section in the results part. 

4) The boundaries of Table 1 extend beyond the page. What is the point of indicating the author of the article and the year in the first column?

Thank you for pointing this out. Indeed, the Table is long; however we expect this issue to be solved during the publication process by changing the Table orientation.

Author and year were included, as in most reviews, to enable the readership to identify the relevant research easily.

5) Subsections of section 3 are numbered as 2.1, etc.

Thank you for highlighting this number inconsistency in section 3. 

We corrected the number sequence in section 3. 

Reviewer 3 Report

Comments and Suggestions for Authors

 A fascinating review exosomes as novel diagnostic and therapeutic agents for non-melanoma skin cancer: As stated, exosomes have an essential role in the pathogenesis and metastasis of cSCC by inducing microenvironmental changes and cellular interaction between cancerous and benign cells. An increasing number of studies focus on exosome-based therapy, reporting potential therapeutic targets. However, further research is needed to clarify the clinical applicability of these findings.

I suggest to add a small paragraph on immunotherapy with cemiplimab and oral therapy with vismodegib and sonidegib

The topic is interesting, English is fine

Author Response

Thank you for reviewing our manuscript and your kind words.

I suggest to add a small paragraph on immunotherapy with cemiplimab and oral therapy with vismodegib and sonidegib

Thank you for pointing this out. A paragraph was added (lines 402-412), as requested.

Round 2

Reviewer 2 Report

Comments and Suggestions for Authors

As the authors of the article wrote in their response to comments, it is impossible to guarantee that the articles with which the authors worked in the process of preparing the manuscript were devoted specifically to exosomes and not to the extracellular vesicles (in general). In such a case, both the title (obviously) and text of the article (in most cases) should use the term sEV unless the authors can be sure that the subject was exosomes complying with MISEV2018 or later edition.

Since we live in 2024, there is no point in reducing the scientific value of the article by explaining that the authors do not have the opportunity to double-check the results. But if the authors read not only the abstracts of the articles, then this work is extremely important for the authors of the REVIEW ARTICLE.

The reviewer has no other comments on the content in its present form.

Author Response

Thank you again for emphasizing the importance of using precise terminology when referring to EV research. As we stated before, our manuscript is a systematic review of the current literature, not an original research article. We point out the nature of our work because our primary objective was to synthetize and analyze the existing studies focusing on what refers as exosomes, even though the term is used in a broader manner than the guidelines recommend.

The decision to retain the use of "exosomes" in our title and throughout the text, was made to accurately capture the full scope of current research. We aim to present the research as it was conducted and reported, respecting the original authors' conclusions. We believe that this choice does not detract from the scientific value of our review but rather highlights the existing challenges, showing the context that the term “exosomes” has been used up to date. Our goal is to accurately portray the included studies, while also discussing these considerations within our manuscript, particularly in the discussion section, where we report the need for greater adherence to guidelines in future research.

We assure you that our review process was thorough, extending to a critical analysis of the methodologies described in full texts.

To further address your concerns though, we have included an additional statement in our manuscript to explicitly acknowledge that, as a review article, our focus is on summarizing and analyzing the existing literature rather than verifying or identifying exosomes according to the strict criteria described in the MISEV guidelines. This clarification will underscore the rationale behind our terminological choices.

We hope this additional explanation addresses your concerns and further clarifies our approach and objectives. 

Reviewer 3 Report

Comments and Suggestions for Authors

Thank you for improving the manuscript

Author Response

Thank you for reviewing our manuscript and your kind words.

Round 3

Reviewer 2 Report

Comments and Suggestions for Authors

Since the data acquired by the systematic review of the current literature contained not only the "exosomes" but also "small extracellular vesicles", etc, the title and content of the article, which uses the more specific term "exosomes", looks completely incorrect. 

Author Response

We appreciate your continued effort to amend our manuscript. After careful reevaluation of your comments, understanding the importance of terminology precision, we revised our systematic review accordingly. To address this, we have made several corrections to ensure that our terminology doesn’t contrast with the current guidelines but also reflects the existing literature. Our initial decision to use the term "exosomes" consistently was based on reflecting the terminology used in the majority of the studies we reviewed. This choice was further supported by instances where despite the general use of broader terms such as EVs or sEVs, specific markers identified as exosomal (e.g., CD63 and CD81) were employed in methodologies such as Western blot techniques, as seen in studies by Overmiller et al. (2017) and Sun et al. (2017). Similarly, Flemming et al. (2020) initially identified sEVs as exosomes but used the term sEVs throughout their study.

Given the main use of the term "exosomes" in the studies reviewed and these instances where studies hint at exosomal characteristics while using broader EV terminology, we chose to maintain consistency with the term "exosomes" in our review, while respecting the original author’s choices where applicable.

We also changed the title of our article to include the term "extracellular vesicles (EVs)" instead of "exosomes." This broader term more accurately encompasses the extracellular vesicles described in the studies we reviewed.

Furthermore, we reviewed the manuscript to identify instances where the term "exosomes" was used and could be replaced by the more inclusive term "EVs", without conflicting with the authors' original choice of terminology. Where supported by the cited literature, those adjustments have been made.

Finally, for transparency, we included a brief clarification note in the results to report how our terminology choices concerning “exosomes” throughout the review coincide with the original studies’ authors.